# IMPROVING THE IMPUTATION OF MISSING DATA WITH MARKOV BLANKET DISCOVERY

**Yang Liu, Anthony C. Constantinou**
Machine Intelligence and Decision Systems (MInDS) Research Group
Queen Mary University of London
`{yangliu, a.constantinou}@qmul.ac.uk`

## ABSTRACT

The process of imputation of missing data typically relies on generative and regression models. These approaches often operate on the unrealistic assumption that all of the data features are directly related with one another, and use all of the available features to impute missing values. In this paper, we propose a novel Markov Blanket discovery approach to determine the optimal feature set for a given variable by considering both observed variables and missingness of partially observed variables to account for systematic missingness. We then incorporate this method to the learning process of the state-of-the-art MissForest imputation algorithm, such that it informs MissForest which features to consider to impute missing values, depending on the variable the missing value belongs to. Experiments across different case studies and multiple imputation algorithms show that the proposed solution improves imputation accuracy, both under random and systematic missingness.

## 1 INTRODUCTION

Dealing with missing data values represents a common practice across different scientific domains, especially in clinical (Little et al., 2012; Austin et al., 2021), genomics (Petrazzini et al., 2021) and ecological studies (Alsaber et al., 2021; Zhang & Thorburn, 2022). It represents a problem that can be difficult to address accurately, and this is because missingness can be caused by various known and unknown factors, including machine fault, privacy restriction, data corruption, inconsistencies in the way data are recorded, as well as purely due to human error.

Rubin (1976) categorised the problem of missing data into three classes known as Missing Completely At Random (MCAR), Missing At Random (MAR) and Missing Not At Random (MNAR). We say data is MCAR when the missingness is purely random, i.e., the missing mechanism is independent of both the observed and unobserved values. On the other hand, data is MAR when the missingness is dependent on the observed values but independent of the unobserved values given the observed values; implying that MAR data can be effectively imputed by relying on observed data alone. Lastly, data is said to be MNAR when it is neither MCAR nor MAR and hence, missingness is dependent on both the observed and unobserved values. While it is tempting to simply remove data rows that contain empty data cells, a process often referred to as list-wise deletion or complete case analysis, past studies have shown that such an approach is ineffective since it tends to lead to poorly trained models (Wilkinson, 1999; Baraldi & Enders, 2010). On this basis, the problem of missingness is typically handled by imputation approaches which estimate the missing values, often using regression or generative models, and return a complete data set.

The imputation algorithms are often classified as either statistical or machine learning methods (Lin & Tsai, 2020). Statistical imputation methods include Mean/Mode, which is one of the simplest methods where the imputation is derived by the mean or mode of the observed values found in the same data column. A more advanced statistical method is the Expectation-Maximization (EM) algorithm (Honaker et al., 2011). EM computes the expectation of sufficient statistics given the observed data at the E-step (Expectation), and then maximizes likelihood at the M-step (Maximization). It iterates over these two steps until convergence, at which point the converged parameters are used along with the observed data to impute missing values. Another statistical al-

gorithm is the one proposed by Hastie et al. (2015), called softImpute, which treats imputation as a matrix completion problem and solves it by finding a rank-restricted singular value decomposition. Multiple imputation is another popular statistical method for handling missing data, and considers the uncertainty of missing values. Some classic multiple-imputation algorithms include the Multivariate Normal Imputation (MVNI) (Lee & Carlin, 2010), Multiple Imputation by Chained Equations (MICE) (Van Buuren & Groothuis-Oudshoorn, 2011), and Extreme Learning Machine (ELM) (Sovilj et al., 2016).

On the other hand, one of the earliest imputation methods that come from the Machine Learning (ML) field include the k-nearest neighbour (k-NN) (Zhang, 2012), which imputes empty cells according to their k-nearest observed data points. A well-established ML imputation algorithm is MissForest (Stekhoven & Bühlmann, 2012), which trains a Random Forest (RF) regression model recursively given the observed data, for every variable containing missing values, and uses the trained RF model to impute missing values. Recently, deep generative networks have also been used for imputing missing data values. Yoon et al. (2018) proposed the Generative Adversarial Imputation Nets (GAIN) algorithm which trains the generator to impute missing data and the discriminator to distinguish original data and imputed data, and was shown to have higher imputation accuracy compared to previous approaches. Other ML techniques used for imputation include the optimal transport (Muzellec et al., 2020), a neural network with causal regularizer (Kyono et al., 2021), and automatic model selection (Jarrett et al., 2022).

All of the aforementioned algorithms assume that all the variables in the data correlate with each other, and use all the variables to impute the missing values. Considering all of the data variables increases the risk of over-fitting, but which can be minimised through L1 and L2 regularization methods often employed by ML algorithms. However, regularization leads to models that tend to lack interpretability and theoretical guarantees of correctness. Because this paper focuses on interpretable models, such as those produced by structure learning algorithms, we shall focus on causal feature selection which maintains interpretability, rather than regularization. This is also partly motivated by Dzulkalnine & Sallehuddin (2019) who showed that using uncorrelated variables to impute missing values not only decreases learning efficiency, but also degrades imputation accuracy. On this basis, it has recently been suggested to include a feature selection phase that prunes off potentially unrelated variables, for each variable containing missing values, prior to imputation (Bu et al., 2016; Liu et al., 2020; Hieu Nguyen et al., 2021).

Relevant studies that focus on feature selection for imputation include the work by Doquire & Verleysen (2012) who used Mutual Information (MI) to measure the dependency between variables. They used a greedy forward search procedure to construct the feature subset, which is an iterative process that constructs feature sets that maximise MI with the dependent variable. Sefidian & Daneshpour (2019) also estimate the dependency between variables using MI, and chose to select a set of variables that increase MI above a given threshold, as the features of a given dependent variable. On the other hand, the algorithm proposed by Dzulkalnine & Sallehuddin (2019) applies a fuzzy Principle Component Analysis (PCA) approach to the complete data cases to remove irrelevant variables from the feature set, followed by a SVM classification feature selection task that returns the set of features that maximise accuracy on the dependent variable. Lastly, evolutionary optimisation algorithms have also been adopted for feature selection in imputation, and include differential evolution (Tran et al., 2018), genetic algorithms (Awawdeh et al., 2022), and particle swarm optimisation (Jin et al., 2022).

Recently, causal information has also been adopted to feature selection for missing data imputation. Kyono et al. (2021) proposed to impute missing values of a variable given its causal parents derived from the weights of the input layer in the neural network. Similarly, Yu et al. (2022) proposed the MimMB framework that learns Markov Blankets (MBs) to be used for feature selection in imputation, which is an iterative process that learns MBs from the imputed data and updates the learned MB after each iteration. Note that while MimMB is related to our work, since we also use MB construction for feature selection, an important distinction between the two is that MimMB combines MBs with imputed data whereas, as we later describe in Section 3, the learning phase of MBs that we propose is separated from imputation, accounts for partially observed variables, and improves computational efficiency.

In this paper, we use the graphical expression of missingness proposed by Mohan et al. (2013), known as *m-graph*, which is a graph that captures observed variables in conjunction with the possible

causes of missingness as parents of the partially observed variables. We first show that an original version of the Grow and Shrink (GS) algorithm by Margaritis (2003) is capable of discovering the MBs in m-graphs containing partially observed variables, when applied to test-wise deleted data. Because this approach relies on CI tests with large conditioning sets, we modify GS such that the number of conditioning sets considered for CI tests is reduced. We provide proof that the modified GS is capable of discovering the MBs of partially observed variables in m-graphs, under the same assumptions as with the original GS. We then propose a new imputation algorithm, which we call Markov-Blanket Miss-Forest (MBMF), that combines the modified GS with the state-of-the-art MissForest (MF) imputation algorithm. We evaluate the effectiveness of MBMF on both synthetic and real-world data sets. The empirical experiments show that MBMF outperforms MF, and other relevant state-of-the-art imputation algorithms, under most experiments.

## 2 PRELIMINARIES

### 2.1 BAYESIAN NETWORK

A Bayesian network $\langle \mathcal{G}, \mathbb{P} \rangle$ is a probabilistic graphical model represented by a Directed Acyclic Graph (DAG) $\mathcal{G} = (\boldsymbol{V}, \boldsymbol{E})$ and associated probability distributions $\mathbb{P}$ over $\boldsymbol{V}$. In DAG $\mathcal{G}$, a *path* is a sequence of distinct nodes such that every pair of nodes is adjacent in $\mathcal{G}$. A node $V_i$ is called a *collider* on a path $p$ if at least two of its neighbouring nodes on $p$ are parents of $V_i$ in $\mathcal{G}$. We denote a node $V_j$ as a *descendant* of $V_i$ if there is a path from $V_i$ to $V_j$ such that all arrowheads of the edges on that path are from $V_i$ to $V_j$. A key concept of DAG is *d-separation*, which defines the conditional independence (CI) between variables in DAG.

**D-separation:** Two variables X and Y are d-separated conditional on a variable set $\boldsymbol{Z}$ if every path between $X$ and $Y$ has a node $W$ that satisfies one of the following two conditions: i) $W$ is not a collider and $W \in \boldsymbol{Z}$, or ii) $W$ is a collider and none of $W$ or its descendant are in $\boldsymbol{Z}$ (Pearl, 1988).

Conditional independence entailed by a given DAG via d-separation is not always equivalent to the conditional independence of the corresponding probability distribution. However, we assume the *Markov* and *faithfulness* conditions described below, where the DAG and the corresponding distribution express the same set of conditional independence.

**Markov condition:** Given a DAG $\mathcal{G}$ over $\boldsymbol{V}$, every variable in $\boldsymbol{V}$ is independent of its non-descendants conditional on its parents.

**Faithfulness condition:** Given a DAG $\mathcal{G}$ over $\boldsymbol{V}$, a probability distribution $\mathbb{P}$ is faithful to $\mathcal{G}$ if and only if the conditional independence relationships in $\mathbb{P}$ are exactly the same as the independences entailed by d-separation in $\mathcal{G}$.

Given the faithfulness condition, a variable is conditionally independent of all the other variables given its MB, which contains all its parents, children and parents of its children. We denote MB of a variable $V_i$ as *MB* $(V_i)$.

### 2.2 MISSINGNESS MECHANISM AS AN M-GRAPH

We adopt the graphical representation of the mechanism of missing data known as *m-graph*, proposed by Mohan et al. (2013), and which makes a slightly stronger assumption on MAR and MNAR scenarios compared to the definition proposed by Rubin (1976). Given a missing data set, the observed variables $\boldsymbol{V}$ can be partitioned into fully observed variables $\boldsymbol{V}^o$ and partially observed variables $\boldsymbol{V}^m$. In an m-graph, there is an auxiliary *indicator variable* $R_i$ for each partially observed variable $V_i \in \boldsymbol{V}^m$ that specifies the missingness of $V_i$, such that $R_i = 1$ when $V_i$ is missing and $R_i = 0$ when $V_i$ is observed. We have MCAR if $\boldsymbol{R} \perp\!\!\!\perp \boldsymbol{V}^o \cup \boldsymbol{V}^m$, MAR if $\boldsymbol{R} \perp\!\!\!\perp \boldsymbol{V}^m \mid \boldsymbol{V}^o$, otherwise MNAR. We denote $\boldsymbol{R_S}$ over a set of variables $\boldsymbol{S}$ as $\boldsymbol{R_S} = \cup_{V_i \in \boldsymbol{S}} \{R_i \mid V_i \in \boldsymbol{V}^m\}$. We also make the Assumption 1 and Assumption 2 for indicator variables, as described below and based on (Mohan et al., 2013).

**Assumption 1.** *No missing indicator variable $R_i$ can be a parent of observed variables or other indicator variables, i.e., $R_i$ could only be a leaf node in m-graph.*

**Assumption 2.** *In an m-graph, no edge can exist from a partially observed variable $V_i$ to its corresponding indicator variable $R_i$.*

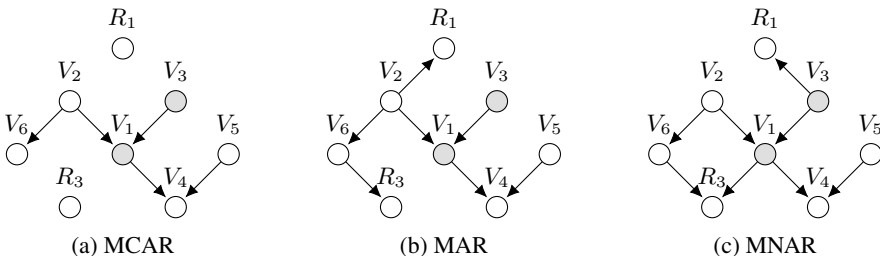

Figure 1: M-graphs under MCAR, MAR and MNAR conditions respectively. Shaded nodes represent partially observed variables.

Given Assumption 1, if two variables $V_i$ and $V_j$ are d-separated by a variable set $\boldsymbol{S}$, they are still d-separated given $\boldsymbol{S} \cup \boldsymbol{R}$. Given Assumption 2, we exclude the situation that there is a causal relation between $V_i$ and $R_i$ in order to avoid performing CI test between $V_i$ and $R_i$. Figure 1 presents three m-graph examples under different mechanisms of missingness.

## 3 MARKOV BLANKET BASED FEATURE SELECTION FOR IMPUTATION

Given the description of the m-graph and causal faithfulness assumption, the problem of feature selection under incomplete data can be converted into a MB discovery problem over m-graphs that contain partially observed variables and missing indicators[1]. Because of the possible causal links between partially observed variables $\boldsymbol{V}^m$ and indicator variables $\boldsymbol{R}$ (i.e., in the case of MNAR), the MB of a partially observed variable is likely to contain both observed and indicator variables. For example, in Figure 1c, the MB of $V_1$ is $\{V_2, V_3, V_4, V_5, V_6, R_3\}$.

We show that the Grow and Shrink (GS) algorithm with test-wise deletion is capable of discovering the m-graph MB of any variable from incomplete data. Here, unlike list-wise deletion which removes all data rows containing at least one missing value, we use test-wise deletion which removes data cases containing missing values in any of the variables involved in the current CI test. The pseudo-code of GS (Margaritis, 2003) is presented in Algorithm 1. Note that we slightly modify the Grow phase, i.e., line 4-6, of GS to eliminate its dependency on the order of the variables in the data (Kitson & Constantinou, 2022).

Given the faithfulness condition, Assumption 1 and Assumption 2, Proposition 1 describes the correctness of GS with test-wise deletion.

**Proposition 1.** *Given the faithfulness condition and Assumptions 1 and 2, for any observed variable $V_i$ in a m-graph $\mathcal{G}$, the output of GS($V_i$, $\boldsymbol{V}^o \cup \boldsymbol{V}^m \cup \boldsymbol{R} - \{V_i, R_i\}$, $D$) is the MB $(V_i)$ in $\mathcal{G}$.*

The proof of Proposition 1 is provided in Appendix A. Therefore, an intuitive way to determine the relevant features for a given variable is to apply the function GS($V_i$, $\boldsymbol{V}^o \cup \boldsymbol{V}^m \cup \boldsymbol{R} - \{V_i, R_i\}$, $D$) on every $V_i \in \boldsymbol{V}^m$. However, this is impractical since the maximum size of the conditioning sets used for CI testing is $|\boldsymbol{V}^o| + 2|\boldsymbol{V}^m| - 3$. In practice, the accuracy of CI tests drops dramatically as we increase the size of the conditional set (Tsamardinos et al., 2003). To address this, we propose the Markov Blanket Feature Selection (MBFS, Algorithm 2) that aims to restrict the maximum size of the conditional set used by CI tests to $|\boldsymbol{V}^o| + |\boldsymbol{V}^m| - 1$.

MBFS involves two phases, where the first phase involves learning the *intrinsic MB* of each partially observed variable. Given a m-graph $\mathcal{G}$, we define the *intrinsic MB* of a variable $V_i$ as the set of variables that are still in the MB of $V_i$ after removing all indicator variables from $\mathcal{G}$. We denote the intrinsic MB of $V_i$ by *iMB* $(V_i)$. Note that *iMB* $(V_i)$ is not necessarily equivalent to the set of observed variables in *MB* $(V_i)$. This is because the missing indicators might be a common effect of two observed variables. For example, the intrinsic MB of $V_1$ in Figure 1c is $\{V_2, V_3, V_4, V_5\}$,

---

[1]To impute the missing values of an incomplete variable, we consider its MB, rather than only its parent variables, for two reasons. Firstly, the parents of an incomplete (or even a complete) variable are not guaranteed to be identifiable from observational data. Secondly, the MB contains the set of nodes that can make the given variable independent over all other variables present in the input data.

---

**Algorithm 1** The Grow and Shrink (GS) algorithm with test-wise deletion

---

    **Input** target variable $X$, candidate variables set $\boldsymbol{S}$, data $D$
    **Output** candidate Markov Blanket *CMB* of $X$

1: **procedure** GS$(X, \boldsymbol{S}, D)$
2:     $CMB \leftarrow \varnothing$
3:     **repeat**                                                    ▷ Grow phase
4:         **if** any $X \not\!\perp\!\!\!\perp S_i \mid CMB, \boldsymbol{R}_{\{X,S_i\}\cup CMB} = \boldsymbol{0}$ for $S_i \in \boldsymbol{S}$ **then**
5:             add $S_i$ with the lowest p-value to *CMB*
6:             remove $S_i$ from $\boldsymbol{S}$
7:         **end if**
8:     **until** *CMB* stays unchanged
9:     **for each** $Y \in CMB$ **do**                                       ▷ Shrink phase
10:        **if** $X \perp\!\!\!\perp Y \mid CMB - \{Y\}, \boldsymbol{R}_{\{X\}\cup CMB} = \boldsymbol{0}$ **then**
11:            remove $Y$ from *CMB*
12:        **end if**
13:     **end for**
14:     **return** *CMB*
15: **end procedure**

---

whereas the standard MB would have also included $V_6$ and $R_3$. It is worth noting that, during phase 1, some nodes that do not belong in *iMB* $(V_i)$ may still be included in the output *CMB*. However, as we show in Appendix B, these nodes are still in the *MB* $(V_i)$ in m-graph. The phase 2 aims to learn all the parents of the missing indicators, in order to complete the feature set of *MB* $(V_i)$. Proposition 2 states that MBFS is capable of learning *MB* $(V_i)$ from missing data for any $V_i \in \boldsymbol{V}^m$ in a m-graph and thus, it could serve as an effective feature selection approach for imputation algorithms.

---

**Algorithm 2** Markov Blanket-based Feature Selection (MBFS)

---

    **Input** partially observed variable $V_i$, data $D$
    **Output** candidate Markov Blanket *CMB* of $V_i$

1: **procedure** MBFS$(V_i, D)$
        ▷ Phase 1 (discover intrinsic MB)
2:     $CMB \leftarrow$ GS $(V_i, \boldsymbol{V}^o \cup \boldsymbol{V}^m - \{V_i\}, D)$
        ▷ Phase 2 (discover other variables in MB caused by indicators)
3:     **for each** $R_j \in \boldsymbol{R} - \{R_i\}$ **do**
4:         $CPS \leftarrow \boldsymbol{V}^o \cup \boldsymbol{V}^m - \{V_j\}$
5:         **for each** $V_k \in CPS$ **do**
6:             **if** $R_j \perp\!\!\!\perp V_k \mid \boldsymbol{S}, \boldsymbol{R}_{\{V_k\}\cup \boldsymbol{S}} = \boldsymbol{0}$ for any $\boldsymbol{S} \subseteq CPS$ **then**
7:                remove $V_k$ from CPS
8:             **end if**
9:         **end for**
10:        **if** $V_i \in CPS$ **then**
11:            $CMB \leftarrow CMB \cup \{R_j\} \cup CPS$
12:        **end if**
13:     **end for**
14:     **return** *CMB*
15: **end procedure**

---

**Proposition 2.** *Given the faithfulness condition, Assumptions 1 and 2, for any observed variable $V_i$ in a m-graph $\mathcal{G}$, MBFS$(V_i, D)$ returns $MB(V_i)$ in $\mathcal{G}$.*

The proof of Proposition 2 is provided in Appendix B. Appendix C discusses the implications on the learning performance of MBFS when the Assumptions 1 and 2 are violated. We then propose a modified version of MissForest that incorporates MBFS as a feature selection process. The modified version of MissForest, which we call Markov Blanket MissForest (MBMF), takes the feature set *MBFS* $(V_i, D)$ for each partially observed variables $V_i$, as opposed to considering all of the other observed variables as the explanatory features of $V_i$ in the Random Forest regression model used in

MissForest. In other words, MBMF accounts for the possible causal relationships between partially observed variables and the missing indicators, to minimise the risk of considering irrelevant observed variables for imputation by MissForest.

## 4 EXPERIMENTS

We test the proposed MBMF algorithm with reference to the standard version of MissForest (MF), the commonly used imputation algorithms Mean and Mode, the K-Nearest Neighbour (KNN), and two state-of-the-art algortihms; the softImpute and GAIN algorithms. While the evaluation includes experiments on both continuous and categorical data, some of the other algorithms can only process one of the two types of input data and hence, their application is restricted to continuous data (Mean and GAIN) or categorical data (Mode). We use the scikit-learn python package (Pedregosa et al., 2011) to test the Mean, Mode and KNN algorithms, the MissForest R package (Stekhoven & Stekhoven, 2013) to test MF, the softImpute R package (Hastie et al., 2015) to test SoftImpute, and the publicly available source code of GAIN. The implementation of MBMF, described in this paper, is available at: https://github.com/Enderlogic/Markov-Blanket-based-Feature-Selection.

MBMF is applied to continuous data using the Pearson's correlation test for CI tests, and to categorical data using the G-test statistic, both of which are the default choices for GS. We also consider the default threshold for independence, which is 0.1 for CI p-value tests. The other algorithms are also tested with their default hyper-parameters as implemented in their corresponding packages listed above.

### 4.1 SYNTHETIC CASE STUDIES BASED ON REAL-WORLD BNS

We first evaluate the algorithms by applying them to synthetic data sampled from three BNs, ECOLI70, MAGIC-IRRI and ARTH150, taken from the bnlearn repository (Scutari, 2010). Details about these graphical networks can be found in Appendix D. We generate complete data sets for each network with sample sizes 500, 1000, 2000 and 3000. Then, for each complete data set, we create nine incomplete data sets composed of different combinations of missingness rates (i.e., 10%, 30% and 50%) and missingness assumptions (i.e., MCAR, MAR and MNAR). Appendix E describes the process we followed to generate data sets with different types of missingness.

### 4.2 EVALUATION PROCESS

The imputation accuracy is evaluated using two different approaches. The first approach involves retrieving the Root Mean Squared Error (RMSE) between imputed data and complete data. Because RMSE is sensitive to the discrepancy between absolute data values, we normalise the complete data column-wise and re-scale the imputed data with the same normalisation parameters to eliminate bias.

The second approach involves assessing the impact of imputation on structural learning accuracy. We do this by comparing the Completed Partially Directed Acyclic Graphs (CPDAGs) learned by the state-of-the-art GES causal structure learning algorithm (Chickering, 2002) from imputed data sets produced by the different imputation algorithms. The second approach is helpful because, while it is reasonable to assume that higher imputation accuracy helps causal machine learning, it is possible that some imputed values are more important than others. Causal structure learning represents a good approach to test this, and we use the F1 score to measure the accuracy of graphical structures learned by GES, as follows:

$$F1 = \frac{2TP}{2TP + FP + FN},$$

where $TP$, $FP$ and $FN$ represent the number of true positive, false positive and false negative edges in the learned CPDAG, relative to the true CPDAG. For more information on how to retrieve the CPDAG of a DAG, please refer to (Chickering, 2002). Readers are also referred to (Kitson et al., 2023) for a review of structure learning.

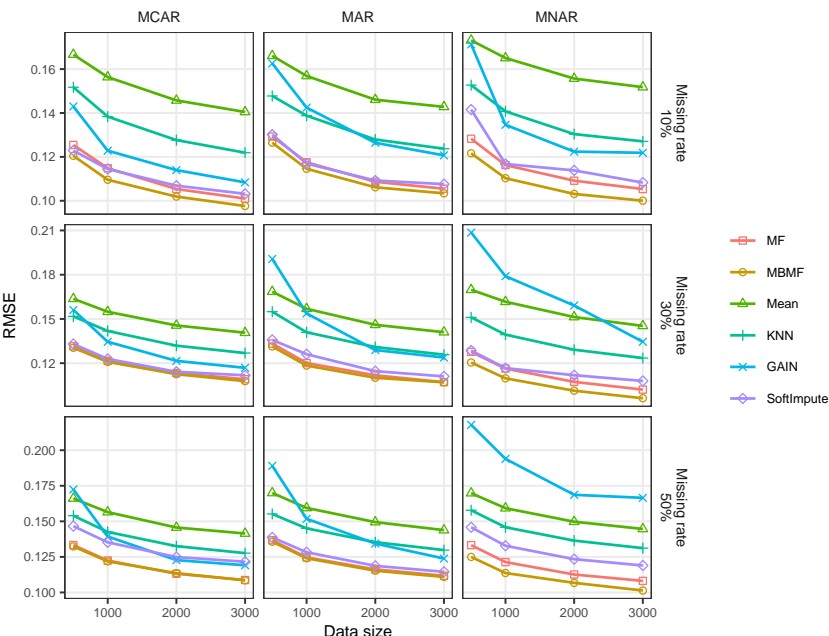

Figure 2: Average RMSE between complete and imputed data produced by the different algorithms. A Lower score represents better performance.

## 4.3 RESULTS

Figure 2 depicts the average RMSE of imputed data produced by the different algorithms under different sample sizes. Note that the results we report on GAIN are, to some degree, inconsistent with the results presented in the original paper (Yoon et al., 2018) , but are consistent with the results presented in follow-up studies (You et al., 2020; Nazabal et al., 2020; Kyono et al., 2021; Jarrett et al., 2022). In general, the proposed MBMF algorithm is found to outperform the baseline MF under all scenarios of missingness. Specifically, for MCAR and MAR scenarios with missing rate 10%, MBMF provides a considerable improvement over MF, but this improvement diminishes with the higher missing rates of 30% and 50%. This is because a higher missing rate tends to decrease the sample size of the test-wise deleted data, which in turn reduces the accuracy of CI tests and the discovered *MB* set by MBFS. Importantly, MBMF outperforms MF considerably under all MNAR settings, which better reflect real-world missingness that is generally systematic. None of the other imputation algorithms provide satisfactory performance in terms of RMSE; at least relative to the MBMF and MF algorithms.

Figure 3 presents the average F1 scores corresponding to the graphs learned by GES, given the imputed data sets produced by the different imputation algorithms, and across the different sample sizes. While this evaluation approach decreases the discrepancy in performance scores between the top performing imputation algorithms, the results are consistent with those presented in Figure 2 since MBMF and MF are found to perform better than the other algorithms in almost all cases, and MBNF performs better than MF in most experiments. Specifically, MBMF and MF produce similar performance when the rate of missingness is lowest at 10%, with their performance being close to that produced with complete data (dashed line). These results serve as empirical evidence that both MFBF and MF imputation algorithms perform exceptionally well with relatively low rates of missingness. When the rate of missingness increases to 30%, MBMF performs better than MF in most cases. However, when the rate of missingness is highest, at 50%, there is no clear winner between MBMF and MF.

Lastly, we evaluate the computational efficiency of MBMF relative to the original MF. As shown in Figure 4, MBMF is generally more efficient than MF. Note that while MBMF involves an additional phase needed to perform feature selection, the additional time spent by MBMF in that phase is countered by the reduced time MBMF spends to actually impute values. This is because MBMF

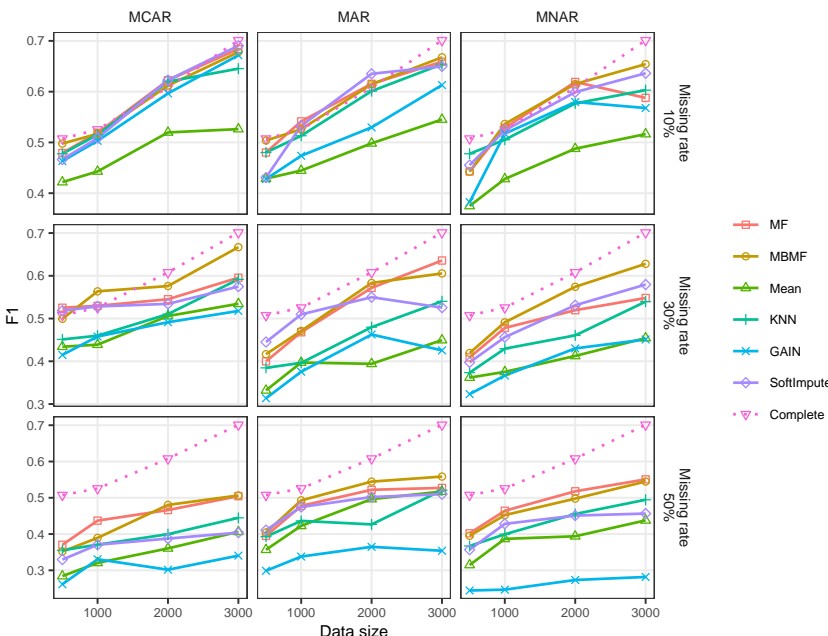

Figure 3: Average F1 scores of the graphs learned by GES from data imputed by the different algorithms. A higher F1 score represents better performance. The dashed line represents the performance of GES when applied to complete data.

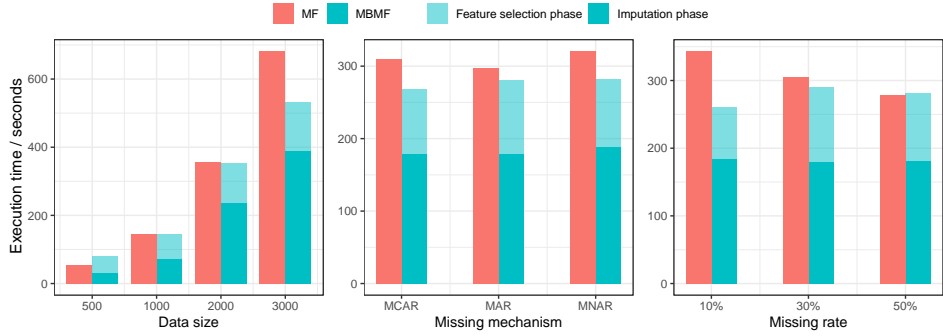

Figure 4: Average execution time of MF and MBMF under different sample sizes, mechanisms of missingness, and rates of missingness.

will almost always consider less features than MF during the imputation phase. Specifically, MBMF is slower than MF at the lowest sample size, but becomes increasingly faster than MF with increasing sample size. Averaging the results across the different mechanisms of missingness and missing rates also shows that MBMF is, in general, more efficient than MF. Note that MF is slightly more efficient when the rate of missingness is at its highest, 50%, since MF trains its RF regression model on observed data rows only; implying that a higher rate of missingness decreases the training data passed to the RF. The impact of the rate of missingness is higher on MF than MBMF since the number of independent features considered by MF is, in general, considerably higher than those considered by MBMF.

## 4.4 REAL-WORLD CASE STUDY

We repeat the evaluation by applying the imputation algorithms to six real-world data sets retrieved from the UCI data repository (Dua & Graff, 2017). A summary of these data sets is given in Table 3 in Appendix F. We simulate missingness using the same strategy as described in Section 4.1. Specif-

|          | Mean/Mode   | KNN         | SoftImpute  | GAIN        | MF          | MBMF        |
|----------|-------------|-------------|-------------|-------------|-------------|-------------|
| Iris     | .284±.085   | .103±.029   | .198±.055   | .164±.060   | .092 ± .026 | **.092±.025** |
| Breast   | .149±.012   | .113±.009   | .140±.040   | .098±.029   | .057 ± .009 | **.054±.009** |
| Wine     | .123±.009   | .109±.013   | .101±.010   | .106±.009   | **.039±.013** | .040 ± .013 |
| Game     | .563±.020   | .501±.072   | -           | -           | .505 ± .064 | **.496±.070** |
| Car      | .675±.074   | .613±.073   | -           | -           | .655 ± .107 | **.593±.061** |
| Mushroom | .443±.062   | .294±.083   | -           | -           | .207 ± .060 | **.203±.058** |

Table 1: Average RMSE and PFC scores, and their standard deviations, for different imputation algorithms and real-world data combinations. Lower RMSE and PFC scores represent better performance.

ically, for each complete real data set, we generate nine incomplete data sets composed of different rates and assumptions of missingness.

We use RMSE to evaluate the imputation accuracy when the algorithms are applied to continuous data, and the Proportion of Falsely Classified entries (PFC) when applied to categorical data. Note that the second evaluation approach, which involves investigating the impact of imputation accuracy on causal structure learning as described in subsection 4.2, is unsuitable here since we do not have a ground truth causal graph.

Table 1 presents the average scores and standard deviation. The results show that MBMF and MF continue to outperform the other algorithms, and that MBMF continues to outperform MF in most of the experiments. On this basis, we conclude that the results obtained from the real-world data sets are consistent with the results obtained from the synthetic data sets.

## 5 CONCLUSION

This paper described a novel feature selection algorithm, called MBFS, that recovers the Markov Blanket of partially observed variables based on the graphical expression of missingness known as the m-graph, which captures observed variables together with missingness indicators and the possible causal links between them. We then incorporated MBFS into the imputation process of MissForest, to formulate a new algorithm suitable for imputation under both random and systematic missingness, which we call MBMF.

Empirical experiments based on both synthetic and real-world data sets show that MBMF outperforms the baseline MissForest in most of the experimental settings, and outperforms considerably other well-known or state-of-the-art imputation algorithms under both random and systematic missingness. Moreover, while MBMF incorporates an additional learning phase needed to perform feature selection for each partially observed variable found in the input data, the results show that MBMF is generally more efficient than the baseline MissForest, especially at larger sample sizes where efficiency matters the most. This is because the time saved during imputation due to prior feature selection is higher than the additional time spent determining the best features for each partially observed variable.

Because the feature selection phase can be independent of the imputation phase (i.e., by using the MBFS algorithm alone), future research works could extend this work to different relevant directions where feature selection is deemed to be important. For example, MBFS could be combined with other imputation algorithms, including those based on deep learning (Mattei & Frellsen, 2019; Fortuin et al., 2020; Lin et al., 2022) which are generally powerful but which tend to be time consuming and to overfit the data, since they typically process large numbers of uncorrelated features. Moreover, since each MB discovered by MBFS could be used to construct the complete m-graph of the input data set, a rather different possible direction for future research would be to investigate the capability of MBFS in recovering the entire m-graph of the input data. This task would require structural rules to deal with collisions between MBs as well as cycles, and would essentially convert MBFS into a structure learning algorithm.

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

## A    PROOF OF PROPOSITION 1

Given Assumption 1 and Assumption 2, $R_i$ is not a child of $V_i$ nor a parent of any other variables. Therefore, we do not need to consider $R_i$ when learning $MB(V_i)$.

We first show that the output Candidate Markov Blanket (*CMB*; i.e., output of Algorithm 1) set learned at the *Grow* phase contains all the variables of $MB(V_i)$. Assume $Y$ is a parent or child of $V_i$, given the faithfulness condition, $V_i \not\perp\!\!\!\perp Y \mid CMB, \boldsymbol{R}_{\{V_i, Y\} \cup CMB} = \boldsymbol{0}$ for any $CMB \subseteq \boldsymbol{V}^o \cup \boldsymbol{V}^m \cup \boldsymbol{R} - \{V_i, R_i\}$. Therefore, $Y$ will always be added to *CMB* during the *Grow* phase. Assume $Z \to Y$ and that $Y$ is also a child of $V_i$. Once $Y$ is added to *CMB*, $V_i$ and $Z$ will not be d-separated by $CMB \cup \boldsymbol{R}_{\{V_i, Y\} \cup CMB}$ and thus, $Z$ will also be added to *CMB*.

Lastly, we show that the *Shrink* phase preserves the variables in $MB(V_i)$ only. Let us assume that $T$ is the first variable in $MB(V_i)$ when the algorithm enters the *Shrink* phase and attempts to remove variables in *CMB*, which have already been added during the *Grow* phase. Because $MB(V_i) - \{T\} \subseteq CMB$, irrespective of $T$ being a neighbour or a parent of a child of $V_i$, we always have $V_i \not\perp\!\!\!\perp T \mid CMB - \{T\}, \boldsymbol{R}_{\{V_i\} \cup CMB} = \boldsymbol{0}$ given the faithfulness condition. Therefore, no variable in $MB(V_i)$ will be removed during the *Shrink* phase. On the other hand, if we assume $T \notin MB(V_i)$, since $MB(V_i) \subseteq CMB \cup \boldsymbol{R}_{\{V_i\} \cup CMB} - \{T\}$, we have $V_i \perp\!\!\!\perp T \mid CMB - \{T\}, \boldsymbol{R}_{\{V_i\} \cup CMB} = \boldsymbol{0}$ given the faithfulness condition and thus, $T$ will be removed from *CMB*.

## B    PROOF OF PROPOSITION 2

For convenience, in this section we denote path $p$ as the path being *blocked* by a variable set $\boldsymbol{S}$ if there is at least one node on $p$ that satisfies either 1) it is a non-collider and in $\boldsymbol{S}$ or 2) it is a collider and neither it nor any of its descendants are in $\boldsymbol{S}$. Thus, if all paths between $V_i$ and $V_j$ are blocked by $\boldsymbol{S}$, $V_i$ and $V_j$ are d-separated by $\boldsymbol{S}$. We firstly show that the *CMB* returned by

GS $(V_i, \boldsymbol{V}^o \cup \boldsymbol{V}^m - \{V_i\}, D)$ (phase 1 of MBFS) contains all but only variables in $iMB(V_i)$, in addition to some other observed variables belonging to $MB(V_i)$.

For the Grow phase in GS $(V_i, \boldsymbol{V}^o \cup \boldsymbol{V}^m - \{V_i\}, D)$, assume that $Y$ is a parent or child of $V_i$, and that given the faithfulness condition, $V_i \not\perp\!\!\!\perp Y \mid CMB, \boldsymbol{R}_{\{V_i, Y\} \cup CMB} = \boldsymbol{0}$ for any $CMB \subseteq \boldsymbol{V}^o \cup \boldsymbol{V}^m - \{V_i\}$. Therefore, $Y$ will be added in $CMB$. Assume that $Z$ is a parent of an observed variable $Y$, which is a child of $V_i$. After $Y$ is added to $CMB$, $V_i$ and $Z$ can no longer be d-separated by $CMB \cup \boldsymbol{R}_{\{V_i, Y\} \cup CMB}$. Thus, $Z$ will also be added to $CMB$ eventually. As a result, all variables in $iMB(V_i)$ will be included in $CMB$ at the end of the Grow phase.

For the Shrink phase, we prove two things: 1) that all the variables in $iMB(V_i)$ will remain in $CMB$, and 2) that all the variables not in the $MB(V_i)$ will be removed from $CMB$.

1. Suppose that $T$ is the first variable in $iMB(V_i)$ that the algorithm attempts to remove from $CMB$, and that $V_i$ and $T$ are connected by either a direct edge or a path through a collider in $iMB(V_i)$. Therefore, $T$ and $V_i$ cannot be d-separated by $CMB \cup \boldsymbol{R}_{\{V_i\} \cup CMB} - \{T\}$, since $iMB(V_i) \subseteq CMB \cup \boldsymbol{R}_{\{V_i\} \cup CMB} - \{T\}$. Then, we have $V_i \not\perp\!\!\!\perp T \mid CMB - \{T\}, \boldsymbol{R}_{\{V_i\} \cup CMB} = \boldsymbol{0}$ given the faithfulness condition.

2. Suppose that $T$ is an observed variable in $CMB$ such that $T \notin MB(V_i)$. Given the faithfulness condition, all we need to prove is that all paths between $V_i$ and $T$ are blocked by $CMB \cup \boldsymbol{R}_{\{V_i\} \cup CMB} - \{T\}$.

   - For a path $p$ between $V_i$ and $T$ composed by observed variables only, there is at least one non-collider node on $p$ that belongs to $iMB(V_i)$, such that $p$ is blocked by $CMB \cup \boldsymbol{R}_{\{V_i\} \cup CMB} - \{T\}$, since $iMB(V_i) \subseteq CMB \cup \boldsymbol{R}_{\{V_i\} \cup CMB} - \{T\}$.

   - For a path $p$ between $V_i$ and $T$ contains indicator variables, when an observed node $V_j$ is adjacent to $V_i$ on $p$, at least one non-collider (either $V_j$ or the parent of $V_j$) on $p$ will also be in $iMB(V_i)$. This is because no indicator variable can be a parent of $V_j$ according to Assumption 1, which implies that $p$ will be blocked by $iMB(V_i)$ as well as $CMB \cup \boldsymbol{R}_{\{V_i\} \cup CMB} - \{T\}$. If the adjacent node of $V_i$ on $p$ is an indicator variable $R_j$, then according to Assumptions 1 and 2, $R_j \neq R_i$ and it must have another parent $V_k \neq T$ on $p$ since $T \notin MB(V_i)$. If $R_j \notin \boldsymbol{R}_{CMB}$, path $p$ is blocked by $CMB \cup \boldsymbol{R}_{\{V_i\} \cup CMB} - \{T\}$. If $R_j \in \boldsymbol{R}_{CMB}$, $V_k$ must be included in $CMB$ during the Grow phase, and cannot be removed during the Shrink phase since $V_i$ and $V_k$ cannot be d-separated by $CMB \cup \boldsymbol{R}_{\{V_i, V_k\} \cup CMB}$. Therefore, $p$ is still blocked by $CMB \cup \boldsymbol{R}_{\{V_i\} \cup CMB} - \{T\}$ since $V_k$ is a non-collider on $p$ in $CMB \cup \boldsymbol{R}_{\{V_i\} \cup CMB} - \{T\}$.

In the phase 2 of MBFS, all the parents of each $R_j$ should remain in $CPS$ as they cannot be d-separated from $R_j$ given any set composed of observed variables. Besides, all of the other variables should be removed from $CPS$ as they can be d-separated from $R_j$ given any set that contains all the parents of $R_j$. As a result, $CMB$ should return $MB(V_i)$ at the end of phase 2.

## C  THE IMPLICATIONS OF VIOLATING ASSUMPTIONS 1 AND 2 OF MBFS

If the Assumption 1 is violated (e.g., when an indicator variable is also a parent of some other variable), then both phases of MBFS may produce redundant variables in the $CMB$ set. Let us suppose that the true m-graph is the one shown in Figure 5. If we use MBFS to learn the MB of $V_1$, then $V_2$ will be included in the $CMB$ set of Phase 1, and this is because $V_1$ and $V_2$ are independent only when conditioning on $R_4$, whereas $V_4$ and $V_1$ are always independent. This implies that MBFS will never test $V_1$ and $V_2$ for CI conditional on $R_4$ during Phase 1. In Phase 2, $V_6$ will be included in the $CPS$ set of $R_5$ and thus, it will also be added in $CMB$ of $V_1$.

If Assumption 2 is violated (e.g., when a partially observed variable $V_i$ is also the parent of its own missingness indicator $R_i$), this will cause MBFS not to include $R_i$ in the $CMB$ of $V_i$, since MBFS would not consider $V_i$ as a candidate variable in the $CPS$ of $R_i$ during Phase 2.

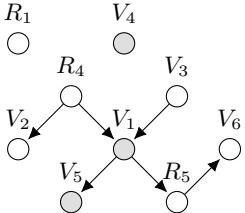

Figure 5: A hypothetical m-graph used to described the implications of violating Assumption 1 of MBFS. Shaded nodes represent partially observed variables.

## D  SUMMARY OF THE REAL-WORLD BNS IN SYNTHETIC EXPERIMENTS

|  | Number of variables | Number of edges | Data type |
|---|---|---|---|
| ECOLI70 | 46 | 70 | Continuous |
| MAGIC-IRRI | 64 | 102 | Continuous |
| ARTH150 | 107 | 150 | Continuous |

Table 2: Summary of the real-world BNs used in Section 4.1.

## E  MISSING VALUE GENERATION PROCESS

We randomly choose 50% of the variables to be made partially observed. We use $\alpha$ to represent the rate of missingness for each of those variables. This process differs depending on the underlying assumption of missingness. Specifically,

1. For the MCAR condition, a value in $V_i$ is removed with probability $P(R_i = 1) = \alpha$.

2. For the MAR condition, a fully observed variable $V_j \in \boldsymbol{V}^o$ is randomly assigned as the parent of $R_i$. For continuous data, we denote the highest 10th-quantile of $V_j$ as $q_j$, and remove values in $V_i$ with the following conditional probabilities: $P(R_i = 1 \mid V_j > q_j) = 2\alpha$, $P(R_i = 1 \mid V_j <= q_j) = \frac{8}{9}\alpha$. For categorical data, we select a set of states $\boldsymbol{v}_j$ of $V_j$ such that $\alpha < P(V_j = \boldsymbol{v}_j) < 1$, then we remove values in $V_i$ with the following conditional probabilities: $P(R_i = 1 \mid V_j \neq \boldsymbol{v}_j) = 2\alpha$, $P(R_i = 1 \mid V_j = \boldsymbol{v}_j) = 2\alpha - \frac{\alpha}{P(V_j = \boldsymbol{v}_j)}$.

3. For the MNAR condition, a partially observed variable $V_j \in \boldsymbol{V}^m - \{V_i\}$ is randomly assigned as the parent of $R_i$. Variable values are then removed for each $V_i$ using the same strategy as in the case of MAR.

## F  SUMMARY OF THE REAL-WORLD DATA SETS

|  | Number of variables | Number of instances | Data type |
|---|---|---|---|
| Iris | 4 | 150 | Continuous |
| Breast | 30 | 569 | Continuous |
| Wine | 11 | 1599 | Continuous |
| Game | 10 | 958 | Categorical |
| Car | 7 | 1728 | Categorical |
| Mushroom | 22 | 8124 | Categorical |

Table 3: Summary of the real-world data sets used in Section 4.4.

