# OpenReview forum: "Improving the imputation of missing data with Markov Blanket discovery"
_ICLR.cc/2023/Conference — ICLR 2023 poster_

### Official Review · Reviewer_p56x · 2022-10-23

**Confidence:** 3
**Clarity, Quality, Novelty And Reproducibility:** The paper is clear. Markov Blanket + …
**Correctness:** 4
**Technical Novelty And Significance:** 2
**Empirical Novelty And Significance:** 2
**Recommendation:** 5

**Strength And Weaknesses:**

Strength:
1. The authors propose a markov blanked based feature selection algorithm for missing data. The selected features are then used in MissForest.

Weakness:
1. The authors should include more recent works to claim that their method is SOTA. MissForest is kind of old (10 years old). More recent SOTA should be included. For example,
MIRACLE (Kyono et al., 2021)
Sinkhorn (Muzellec et al., 2020)
SoftImpute (Hastie et al., 2015)
HyperImpute (Jarrettet al., 2022)
2. MimMB is similar to MBMF. Empirical comparison should also be included.
3. Unlike MimMB, the authors claim that improves computational efficiency. A comparison of execution time is more persuasive. If it's trivial, the authors can correct me without making this new result of computational efficiency.

**Summary Of The Paper:**

The authors propose a feature selection method based on markov blanket under missing data regime and apply missforest after feature selection. The main contribution is the feature selection algorithm.

**Summary Of The Review:**

I think this paper is at the borderline. The authors do not compare to many new SOTA methods, which I believe have better results than MissForest. But it can be the reason that MissForest is not that good so MBMF is not as good as those SOTA methods.

---

> ### Author Response · Authors · 2022-11-16
> **Response to Reviewer p56x**
>
> Thank you for your time on reviewing our submission. We respond to your comments in the same order:
> * Regarding including results against a SOTA algorithm: Please note we already report results against the GAIN algorithm which is SOTA. To address your comment, we have now extended the results and text to include one more SOTA algorithm from those suggested (SoftImpute). We now also reference the algorithms that you mentioned, in introduction.
> * On adding MimMB in the empirical comparisons: We are sorry to report we have been unable to do this since the authors of MimMB have not made their full source-code publicly available (they only released their feature selection code but not the full code that combines their feature selection process with the imputation process).
> * On computational efficiency versus MimMB: Recall that MimMB is an iterative process, and in each iteration, MimMB uses an imputation algorithm to impute incomplete data and learns MB for each partially observed variable based on the latest imputed data. This means that MimMB needs to execute an imputation algorithm, such as MF, multiple times. As shown in Figure 4, when the sample size is equal to or larger than 1000, MBFS becomes faster than MF. This means that MBMF will be many times faster than MimMB, when MinMB relies on MF for imputation.

---

### Official Review · Reviewer_14vZ · 2022-10-24

**Confidence:** 4
**Clarity, Quality, Novelty And Reproducibility:** See below.
**Correctness:** 4
**Technical Novelty And Significance:** 3
**Empirical Novelty And Significance:** 2
**Recommendation:** 8

**Strength And Weaknesses:**

See below.

**Summary Of The Paper:**

The paper proposes a method for feature selection to be used prior to estimating an imputation model. The method is based on finding the Markov blanket for each partially observed variable. In synthetic and semi-synthetic experiments, the methods slightly outperforms an imputation model trained without feature selection with a lower total run time (feature selection + model estimation).

**Summary Of The Review:**

Overall, I found this paper exceptionally clear and well-written. The method is interesting and the experiments are reasonable and well-executed. My only overall concern is regards motivation. By and large, ML methods favor general regularization over using feature selection as a preprocessing step, so I was a bit unconvinced by the initial motivation that feature selection is necessary to improve imputation performance. I think this intuition is born out by the experiments which seem to demonstrate only a small improvement over the base method applied without feature selection. I think the intro would be improved with a bit more justification for the approach and, specifically, why the authors believe feature selection is the right approach here when regularization has worked so well in other settings. With that said, I still think the approach is reasonable and merits publication.

Minor comments:
1. Overall, I thought the background provided in the intro was excellent. My only quibble is that, in my experience, the most commonly used approach to missing data in statistical settings is multiple imputation and it might be worth mentioning this approach in the intro. Note that improved imputation models mean improve MI.

2. Intro: Modem --> Mode

3. Figures 1 and 2: I think these figures could be cleaned up by organizing them in a 3x3 grid. If you are concerned about space, I recommend sharing x and y axes within the grid.

---

> ### Author Response · Authors · 2022-11-16
> **Response to Reviewer 14vZ**
>
> Thank you for your time on reviewing our submission. We respond to your comments in the same order:
>
>  * On regularization: Although regularization is widely used in many machine learning settings, it offers little interpretability and no theoretical guarantees, both of which are needed when working with causal models or when model interpretability is paramount (which is the case in this paper since we focus on causal structure learning). We have revised the introduction to make the justification clear, as suggested.
> * On adding multiple imputation in intro: This is now done.
> * Intro: Modem $\rightarrow$ Mode typo corrected.
> * Regarding Figures 1 and 2: We have reformatted these figures as suggested.

---

### Official Review · Reviewer_X4h7 · 2022-10-31

**Confidence:** 4
**Correctness:** 3
**Technical Novelty And Significance:** 3
**Empirical Novelty And Significance:** 3
**Recommendation:** 8

**Clarity, Quality, Novelty And Reproducibility:**

The paper was quite well written. However, I did not find any instructions or leads to reproduce the authors' results.

**Strength And Weaknesses:**

STRENGTHS

* Clearly, causality and structure are important elements in imputation, particularly in the tabular domain. Imputation is an important research area that should receive more attention than it currently does.

* I found the paper quite well written. Also, I find it interesting that the authors base their method on a tried and tested algorithm from 2003.

WEAKNESSES

I think the paper was interesting and have only a few questions I would like the authors' opinion on.

* What exactly is the motivation for using the Markov blanket instead of the parents of X_i? Kyono et al. (2020) seem to be doing well with the covariate's parents? (Note that an m-graph is _causal_, saying it is recovered allows assuming that the parents of a variable are _actually_ the causes).

* Would it be useful to try MBMF with other algorithms (beyond MissForest)? If so, why haven't you done this and could you? If not, why not?

* It seems GAIN is not performing very well, despite it being considered the sota, could you elaborate or reason why this is the case?



**Summary Of The Paper:**

The authors propose MBMF, essentially a feature selection procedure for a second step imputation method. With MBMF the authors aim to recover the m-graph that governs a system's missingness (in the tabular domain). Using the m-graph, the authors recover a variable of interest's markov blanket which are used as features for imputation. The second step uses the selected features to impute a missing variable.

Their experiments show that MBMF combined with MissForest outperforms the most adopted imputation strategies.

**Summary Of The Review:**

I think the paper is well written, interesting, and touches on some interesting ideas to marry causal discovery and imputation. I was left with a few questions I hope the authors can answer in their rebuttal.

---

> ### Author Response · Authors · 2022-11-16
> **Response to Reviewer X4h7**
>
> Thank you for your time on reviewing our submission. We answer your questions in the same order:
>
> * On why we use Markov Blanket and not just the parents of a given variable: Because the parents of a variable cannot fully explain the observed outcomes of the given variable. The MB is needed to fully explain the observed distribution. This is because in probabilistic graphical models, inference propagates from parent-to-child, but also from child-to-parent. Moreover, there is no guarantee that the parents of $V_i$ can be identified purely from observational data, and Kyono et al. (2020) did not provide such theoretical guarantee in their paper. In this paper, we can only guarantee that the Markov Blanket of the observed variable $V_i$ can be recovered as long as the assumptions are not violated. Moreover, non-parents in a MB can provide useful information about $V_i$ since $V_i \\not\\!\\perp\\!\\!\\!\\perp MB\\{V_i\\}- Pa\\{V_i\\}\\mid Pa\\{V_i\\}$. Therefore, the MB of $V_i$ is expected to provide higher imputation accuracy compared to the parents of $V_i$.
>
> * On why our algorithm uses MissForest and not some other algorithm: This is because MissForest has been well-documented in the literature, and continues to produce exceptional performance despite being a rather old algorithm. Many new algorithms in this area tend to be based, or partly based, on MissForest. Yes, it is possible to combine our feature selection strategy with some other learning strategies to create additional imputation algorithms. We discuss this in conclusions as a possible direction for future research.
>
> * On why GAIN is not performing very well: Although, the results we present on GAIN may not be consistent with the original paper, but are consistent with multiple follow-up studies. For example, our results are consistent with the four papers listed below which also use GAIN for benchmarking. Please note that we carried further tests on GAIN, including using the data sets that come with its own source code, and we can confirm that results obtained on those data sets are consistent with those presented in our paper.
>
>     You, J., Ma, X., Ding, Y., Kochenderfer, M.J. and Leskovec, J., 2020. Handling missing data with graph representation learning. Advances in Neural Information Processing Systems, 33, pp.19075-19087.
>
>     Nazabal, A., Olmos, P.M., Ghahramani, Z. and Valera, I., 2020. Handling incomplete heterogeneous data using vaes. Pattern Recognition, 107, p.107501.
>
>     Kyono, T., Zhang, Y., Bellot, A. and van der Schaar, M., 2021. MIRACLE: Causally-Aware Imputation via Learning Missing Data Mechanisms. Advances in Neural Information Processing Systems, 34, pp.23806-23817.
>
>     Jarrett, D., Cebere, B.C., Liu, T., Curth, A. and van der Schaar, M., 2022, June. Hyperimpute: Generalized iterative imputation with automatic model selection. In International Conference on Machine Learning (pp. 9916-9937). PMLR.
>
> * Regarding comment "I did not find any instructions or leads to reproduce the authors' results": We have revised the Github repository reported in Section 4, page 6 to include all the necessary files and instructions needed to reproduce the results we report in this paper.

---

> > ### Comment · Reviewer_X4h7 · 2022-11-18
> > **Thank you for your response**
> >
> > I thank the authors for their clarifications.
> >
> > 1. I appreciate that learning the causal parents is not trivial from an observational dataset. Though, one should be careful with terminology. For example, the use of m-graphs actually implies that the graphical model presented is indeed causal. Assuming faithfulness, the set of structural equations associated with the m-graph, would imply that the parents are indeed enough to infer a child variable.
> >
> > I understand that pragmatism warrants the use of the Markov blanket (which I believe wouldn't hurt in case of recovering the actual causal m-graph). I believe a footnote of some kind in light of the above would be a good addition to the final paper.
> >
> > 2. Fair enough.
> >
> > 3. Basically the same argument as above. I believe conflicting results should be acknowledged somewhere. I tend to believe the authors on this (seeing as they provide multiple citations where this is also reported).
> >
> > 4. Thank you
> >
> > I have revised my score to reflect the authors' rebuttal.

---

> > > ### Author Response · Authors · 2022-11-18
> > > **Response to Reviewer X4h7**
> > >
> > > Thank you for your response. We have made two further minor revisions to adopt your suggestions.
> > >
> > > * We added the following new footnote on page 4 to explain why we use MB rather than only the parents.
> > >
> > > > To impute the missing values of an incomplete variable, we consider its MB, rather than only its parent
> > > variables, for two reasons. Firstly, the parents of an incomplete (or even a complete) variable are not guaranteed
> > > to be identifiable from observational data. Secondly, the MB contains the set of nodes that can make the given
> > > variable independent over all other variables present in the input data.
> > >
> > > * We added the following new sentence on page 7 to acknowledge the conflicting results.
> > >
> > > > Note that the results we report on GAIN are, to some degree, inconsistent
> > > with the results presented in the original paper (Yoon et al., 2018) , but are consistent with the results
> > > presented in follow-up studies (You et al., 2020; Nazabal et al., 2020; Kyono et al., 2021; Jarrett
> > > et al., 2022)

---

### Official Review · Reviewer_HRJ6 · 2022-11-04

**Confidence:** 3
**Clarity, Quality, Novelty And Reproducibility:** I think the submission is well written.
**Correctness:** 3
**Technical Novelty And Significance:** 3
**Empirical Novelty And Significance:** 2
**Recommendation:** 8

**Strength And Weaknesses:**

Strengths
- Markov Blanket-based Feature Selection (MBFS) and imputation is interesting
- Experimental results show that MBMF can provide promising imputation accuracy

Weaknesses: I think the paper is interesting and I have a few questions.
- Assumption: Is the assumption 1 "R_i could only be a leaf node in m-graph" made to make the problem of finding Markov Blankets easier? Or is it a theoretical guarantee?
- M-graphs: For each data set, can MBFS recover the m-graph which informs the missingness mechanisms? For each given incomplete data set, can the output of MBFS suggest which underlying assumption of missingness was used to create that incomplete data set?
- Scalability: Experiments are done on data sets with small and moderate sizes (at most 30 variables and at most 8124). Can you comment on the scalability of MBFS?

**Summary Of The Paper:**

This submission proposes a new imputation algorithm called Markov-Blanket Miss-Forest (MBMF), which involves two phases: Markov Blanket-based Feature Selection (MBFS), and MissForest (MF) imputation (Stekhoven & Buhlmann, 2012).

For each partially observed variable V_i, MBFS finds its Markov Blanket within some m-graph which is referred to as the graphical expression of missingness proposed by Mohan et al. (2013).  Then, elements of the Markov Blanket are considered as explanatory features of V_i in the Random Forest regression model used in MF.

Some experiments are conducted to assess the usefulness of MBMF, in comparison with the original MF, and simpler imputation algorithms (Mean, Mode and KNN algorithms).

**Summary Of The Review:**

I think "MBFS" & "imputation algorithm" is an interesting imputation approach and can be implemented with different imputation algorithms. I gave a few questions in "Strength And Weaknesses".

---

> ### Author Response · Authors · 2022-11-16
> **Response to Reviewer HRJ6**
>
> Thank you for your time on reviewing our submission.
>
> Please kindly note that the initial version of the paper already included a SOTA algorithm in benchmarks (the GAIN algorithm), and that the revised paper now includes comparisons against one more SOTA algorithm (called softImpute), based on the suggestions made by another reviewer.
>
> We answer your questions in the same order:
>
> * Regarding the question on Assumption 1: The Assumption 1 is a theoretical guarantee of the proposed algorithm. As stated in Proposition 2 (proof in Appendix B), the MBFS algorithm described in this paper guarantees to return the true Markov Blanket (MB) of an observed variable as long as the underlying assumptions are true. As it is the case with all algorithms, if an assumption is violated, this would decrease the accuracy of the MBs produced by MBFS. We now discuss the implications of this in the new Appendix C.
>
> * Regarding the question on M-graphs: MBFS is a local MB discovery algorithm. The aim of this algorithm is to discover both the observed and indicator variables in the MB of a given incomplete variable. While each MB discovered by MBFS could be used to construct the complete m-graph and detect the underlying missingness assumption, e.g., if any indicator variable is in the MB of a partially observed variable, then the data is supposed to be MNAR, this is out of the scope of this paper since the aim is to perform feature selection on each incomplete variable to impute its missing values, rather than structure learning on the entire data set (which would invalidate the motivation for feature selection). However, we agree that there is merit in considering the recovery of the entire graph, and we now discuss this as a possible direction for future research in conclusions.
>
> * Regarding the question on Scalability: The synthetic experiments already include data sets that contain up to 107 variables (refer to data set ARTH150). The new Appendices D and F now provide details about the networks and data sets used in the synthetic and real experiments to make this clear. Moreover, the results in the first plot of Figure 4 show that MBMF is generally more efficient than the original MF when the sample size reaches 1000, and this improvement in efficiency increases with sample size.

---

> > ### Comment · Reviewer_HRJ6 · 2022-11-21
> > **Thank you for the clarifications and revisions**
> >
> > Thank you for the clarifications and revisions. My concerns was addressed/clarified/discussed. My score was revised accordingly.

---

### Author Response · Authors · 2022-11-16
**Summary of changes in our paper**

We would like to thank all reviewers for their valuable feedback and suggestions. We summarize the changes made in the paper as follows:

* The introduction section now also covers statistical multiple imputation as suggested by reviewer 14vZ, as well as recent deep learning-based imputation methods as suggested by reviewer p56x. We now also better highlight the motivation for this paper.
* Reviewer 14vZ suggested a few other SOTA algorithms we could have considered for benchmarking purposes. We have added the softImpute algorithm to our results and revised text and Figures 2 and 3 accordingly.
* We have extended appendices to include a discussion on what happens to the learning performance of MBFS when its underlying assumptions are violated (this was a question raised by reviewer HRJ6). The appendices now also include further details about the networks and data sets used in the synthetic and real experiments.Due to the page limit, we have moved the proof of Proposition 1 and the description of the missing value generation process to appendices A and E respectively.

---

### Decision · Program_Chairs · 2023-01-20

**Decision:**

Accept: poster

**Justification For Why Not Higher Score:**

Reviewers considered the work well done, but topic and scope might not attract all the general audience.

**Justification For Why Not Lower Score:**

One could claim that the simplicity and scope are reasons for a lower score. However, all reviewers agreed on the validity of the topic and direction. The only really major concern was regarding expanding the comparison to more SOTA ideas, which the authors have put an effort to improve.

**Metareview: Summary, Strengths And Weaknesses:**

Simple paper about Markov blanket feature selection for imputation with the missforest idea. Empirical results are satisfactory, showing superiority at times to some SOTA. The topic might be seen as niche, but reviewers appreciated the work on this topic, indicating that it requires more attention that it gives.

**Note From Pc:**

if the above contains the word "oral" or "spotlight" please see: "oral" presentation means -> notable-top-5% and "spotlight" means -> notable-top-25%. As stated in our emails, we are disassociating presentation type from AC recommendations